# Host Species Affects Bacterial Evenness, but Not Diversity: Comparison of Fecal Bacteria of Cows and Goats Offered the Same Diet

**DOI:** 10.3390/ani12162011

**Published:** 2022-08-09

**Authors:** Tiziana Maria Mahayri, Kateřina Olša Fliegerová, Silvana Mattiello, Stefania Celozzi, Jakub Mrázek, Chahrazed Mekadim, Hana Sechovcová, Simona Kvasnová, Elie Atallah, Giuseppe Moniello

**Affiliations:** 1Laboratory of Anaerobic Microbiology, Institute of Animal Physiology and Genetics, Czech Academy of Science, 14220 Prague, Czech Republic; 2Department of Veterinary Medicine, University of Sassari, 07100 Sassari, Italy; 3Department of Agricultural and Environmental Sciences—Production, Landscape, Agroenergy, University of Milan, 20133 Milan, Italy; 4Department of Microbiology, Nutrition and Dietetics, Czech University of Life Sciences in Prague, 16500 Prague, Czech Republic

**Keywords:** bacterial community, bacterial diversity, fecal bacteria, high-throughput sequencing, ruminant species, diet, cows, goats

## Abstract

**Simple Summary:**

Comparison of bacterial diversity and composition of feces from cows and goats offered the same pasture-based diet revealed that the animal species had no effect on bacterial species richness and diversity, but significantly affected species evenness. Both diet and host species influence the gut microbiome.

**Abstract:**

The aim of this study was to compare the diversity and composition of fecal bacteria in goats and cows offered the same diet and to evaluate the influence of animal species on the gut microbiome. A total of 17 female goats (Blond Adamellan) and 16 female cows (Brown Swiss) kept on an organic farm were fed pasture and hay. Bacterial structure in feces was examined by high-throughput sequencing using the V4–V5 region of the 16S rRNA gene. The Alpha diversity measurements of the bacterial community showed no statistical differences in species richness and diversity between the two groups of ruminants. However, the Pielou evenness index revealed a significant difference and showed higher species evenness in cows compared to goats. Beta diversity measurements showed statistical dissimilarities and significant clustering of bacterial composition between goats and cows. Firmicutes were the dominant phylum in both goats and cows, followed by Bacteroidetes, Proteobacteria, and Spirochaetes. Linear discriminant analysis with effect size (LEfSe) showed a total of 36 significantly different taxa between goats and cows. Notably, the relative abundance of *Ruminococcaceae* UCG-005, *Christensenellaceae* R-7 group, *Ruminococcaceae* UCG-010, *Ruminococcaceae* UCG-009, *Ruminococcaceae* UCG-013, *Ruminococcaceae* UCG-014, *Ruminococcus* 1, *Ruminococcaceae* UCG-002, *Lachnospiraceae* NK4A136 group, *Treponema* 2, *Lachnospiraceae* AC2044 group, and *Bacillus* was higher in goats compared to cows. In contrast, the relative abundance of *Turicibacter*, *Solibacillus*, *Alloprevotella*, *Prevotellaceae* UCG-001, *Negativibacillus*, *Lachnospiraceae* UCG-006, and *Eubacterium hallii* group was higher in cows compared with goats. Our results suggest that diet shapes the bacterial community in feces, but the host species has a significant impact on community structure, as reflected primarily in the relative abundance of certain taxa.

## 1. Introduction

Ruminants are economically important livestock [1] because of their unique ability to convert human-indigestible plant biomass into food products for people, especially milk and meat [2,3,4]. It is well known that ruminants cannot produce enzymes necessary for the decomposition of structural plant polysaccharides and they are dependent on a rich rumen consortium of anaerobic microorganisms to ferment animal feed [5]. Bacteria, protozoa, and fungi involved in the degradation of fiber and other dietary components produce volatile fatty acids, which are the main source of energy for the host [6]. Great attention is paid to the study of rumen microorganisms because their composition affects productivity, the quality of meat and dairy products, and the health status of animals. The diversity and structure of the rumen microbiota is influenced by several factors, including diet [7,8,9,10,11,12,13], ruminant species [7,14,15,16], age [17,18,19,20], geographic location [21], type of production system, and host genotype [22,23,24]. The influence of each factor cannot be precisely quantified, but studies performed on large numbers of animals suggest that diet is a critical factor in shaping the rumen microbial ecosystem [7,25]. 

The majority of studies are performed on rumen content samples, as this part of the forestomach is responsible for feed fermentation. Rumen cannulation, stomach tubing or rumenocentesis are the three main techniques used to collect samples to study ruminal fermentation and microbial community composition [26,27]. These methods are, however, invasive, and not always applicable on non-experimental farms. On the other hand, collection of feces is a non-invasive, simple, and inexpensive procedure [28]. Even if bacterial communities observed in the feces do not reflect those reported in the rumen digesta [29,30,31,32], the fecal community is also influenced by changes in diet [33,34,35,36], and therefore may indicate differences related to other factors, such as animal breed [33,37,38] and age [39]. However, information about the influence of host species on fecal bacterial ecosystems remains very scarce. Some authors described increased bacterial diversity in feces compared to rumen [34,40], but phylogenetic differences between ruminal and fecal bacteriome were dependent on animal species, diet, and experimental design [33,40].

Indeed, animal species has influence on host microbial diversity in the digestive tract. Domestic herbivores have different ingestion and grazing behaviors. For example, cattle are known to be typical grazers [41], whereas goats are known to be browsers and mixed feeders [42]. They have different abilities to exploit plant resources, leading to distinct productive responses. Consequently, the efficient utilization of feed varies among ruminants [14]. Although some differences among species in ruminal digestion and fermentation characteristics have been studied [7,14,15,16], there is still a lack of knowledge regarding the differences in the diversity and community composition of fecal microbial populations related to animal species [43]. As there is a clear need for less invasive sampling methods of ruminants to help relate the microbial population to functional traits [44] and study the factors influencing the diversity and composition of these populations, we believe that fecal samples could be used for this type of analysis in order to determine if the fecal microbiome is also affected by host species.

In this study, we investigated and compared the diversity and structure of the bacterial community in the feces collected from grazing cows and goats on an organic farm. Both groups of animals were fed the same high-fiber diet (pasture and hay), kept in the same location (Ceto, Italy), and both breeds were housed together. These circumstances of animal husbandry provide suitable conditions for elucidating the extent to which the digestive tract microbiome is influenced by host animal species. The fecal bacteriome was examined by high-throughput sequencing (HTS) of 16S rRNA fragments and evaluated for its diversity and taxonomic composition. Based on literature data, we hypothesize that the bacterial composition of samples from both ruminant species will be very similar due to the same diet, but we also expect to observe some variations in their fecal bacterial ecosystems related to the different eating habits and physiology of cows and goats. 

## 2. Materials and Methods

### 2.1. Animals and Sample Collection 

The samples for this study were obtained from 33 animals, 17 female goats (Blond Adamellan), and 16 female cows (Brown Swiss), kept on an organic farm in Ceto (Lombardy, Italy; latitude: 46°22′00″; longitude: 10°21′09″). The farm is private, and the owners gave permission to collect fecal samples on 5 February 2021. The cows and goats were kept separately on the same farm. Goats were in freestalls, whereas cows were in tie stalls. They had free access to separated pastures, which varied according to the availability from morning to afternoon (7:30–17:00), and free access to water, both outside and indoors. The botanical composition of the sward includes many different species and varies mainly depending on the altitude (from 400 to 2000 m a.s.l.), ranging from pastures dominated by *Festuca* spp. to pastures where *Carex* spp. and *Sesleria* spp. are the main botanical species. The animals received polyphyte hay and alfalfa (50% and 50%) ad libitum, 3 times daily, at 7:00/7:30, 17:00, and 18:00/18:30. All the animals were submitted to an eprinomectin-based external treatment against parasites (0.5 mg/kg liveweight). The characteristics of the animals are listed in Appendix A. Fresh fecal samples were collected immediately after defecation while the animals were grazing outdoors in an appropriate way, avoiding bedding contamination. They were placed in a sterile bag and transported in a small portable refrigerator to the Department of Agricultural and Environmental Sciences (Milan, Italy). They were frozen, then a representative aliquot of each sample was freeze-dried (Brizio Basi BVL2, Milan, Italy) and moisture and dry matter content were determined by weighing before and after lyophilization. The dried samples were transported to the Institute of Animal Physiology and Genetics of the Czech Academy of Sciences (Prague, Czech Republic) for further analysis.

### 2.2. DNA Extraction 

Dry feces were disrupted with a mortar and pestle in liquid nitrogen and a sample amount equivalent to approximately 300 mg wet weight was used for nucleic acid isolation. Genomic DNA was extracted using the DNeasy^®^ Plant Pro Kit (Qiagen, Hilden, Germany) according to the manufacturer’s instructions. The concentration and quality of the nucleic acids (ratio 260/280) were checked using a NanoDrop 2000c UV-Vis spectrophotometer (Thermo Scientific, Wilmington, DE, USA), and the DNA extracts were stored at −20 °C until needed for analysis. 

### 2.3. PCR Amplification and Purification 

The DNA isolated from each sample were diluted 10-fold in nuclease-free H_2_O and 2 µL of the diluted DNA solutions were used as templates for the PCR reaction. The bacterial variable V4–V5 region of 16S rRNA was amplified using the specific primer pair BactBF (GGATTAGATACCCTGGTAGT) and BactBR (CACGACACGAGCTGACG) [29]. The PCR reaction was performed using EliZyme^TM^ HS FAST MIX Red Master Mix (Elisabeth Pharmacon, Brno, Czech Republic). Thermal cycling conditions included an initial denaturation for 5 min at 95 °C, followed by 25 cycles consisting of 30 s at 95 °C, 30 s at 57 °C, 30 s at 72 °C, and a final elongation step at 72 °C for 5 min. The length and quality of the amplicons were checked by agarose gel electrophoresis (1.5%) and the PCR products were purified using the Monarch^®^ PCR & DNA Cleanup Kit (New England BioLabs, Ipswich, MA, USA).

### 2.4. Library Perpetration and High-Throughput Sequencing 

The library preparation was performed using the NEBNext Fast DNA Library Prep Set for Ion Torrent (New England BioLabs, Ipswich, MA, USA) and the Ion Xpress Barcode Adapters 1-96 Kit (Thermo Fisher Scientific, Waltham, MA, USA). The length of the target amplicons of DNA libraries was analyzed using the 2100 Bioanalyzer Instrument (Agilent Technologies, Santa Clara, CA, USA). The amplicons were pooled in equimolar ratios based on concentration determined using a KAPA Library Quantification Kit (KAPA Biosystems, Roche, Pleasanton, CA, USA). The template amplification and enrichment were performed by emulsion PCR in the Ion OneTouch™ 2 instrument using the Ion PGM^TM^ HiQ^TM^ View OT2 Kit-400 (Thermo Fisher Scientific, Waltham, MA, USA). The enriched template was sequenced with the Personal Genome Machine (PGM™) System (Thermo Fisher Scientific, Waltham, MA, USA) using the Ion PGM™ Hi-Q™ View Sequencing solutions kit and the Ion 316™ Chip v2 BC according to the manufacturer’s protocols. 

### 2.5. Bioinformatic Analysis 

The raw sequencing reads were first filtered using Ion Torrent software to remove low-quality and polyclonal sequences. The bacterial 16S rRNA gene sequences were retrieved in FASTQ format and analyzed using Qiime2 version 2020.2 software [45]. The sequences were quality filtered, trimmed, and denoised using DADA2, and chimeras were removed [46]. To allow for uniform sampling depth, the dataset was subsampled to a minimum of 2300 reads per sample. The rarefaction curves reached a plateau, indicating that the sequencing depth was sufficient and all the species in the samples were adequately covered (Appendix A). The high-quality sequences were then clustered into Amplicon Sequence Variants (ASVs) using VSEARCH and the taxonomic assignment was performed using a BLAST search against the SILVA database (version 132) with a 97% threshold [47]. The bacterial diversity was assessed using alpha diversity indices such as Chao1, Observed ASVs, Faith’s Phylogenetic Diversity, Pielou Evenness, and Shannon Entropy. Beta diversity was calculated using different distance matrices (weighted and unweighted, UniFrac, Bray-Curtis and Jaccard). The principal coordinate analysis (PCoA) was used to visualize the community associations, and the results were plotted using EMPeror [48]. Linear discriminant analysis (LDA) with an effect size (LEfSe) algorithm [49] was accomplished using the Galaxy web module (http://huttenhower.sph.harvard.edu/galaxy/ (accessed on 20 October 2021) to identify the key phylotypes of the differentially abundant taxa. Sequence information was deposited in the Sequence Read Archive under accession number PRJNA826341.

### 2.6. Statistical Analysis

The Alpha diversity between two groups of animals was compared with a nonparametric test using the Kruskal-Wallis H test. Beta diversity was assessed using permutational multivariate analysis of variance (PERMANOVA) with 999 permutations. In addition, the PERMDISP test was performed to test the homogeneity of dispersions among the animal groups. The detection of taxa with significant differences in abundance between cows and goats was performed using the factorial Kruskal-Wallis test and the pairwise Wilcoxon test. The LEfSe analysis was performed with the following parameters: α = 0.05 and a minimum LDA score = 2.0. 

## 3. Results

### 3.1. Alpha and Beta Diversity 

The bacterial community structure in the feces of cows and goats kept on a pasture-based farm was qualitatively and quantitatively analyzed for species richness, evenness, and phylogenetic diversity. The analysis revealed a lower diversity in the samples from goats. However, with the exception of Pielou’s evenness index (a measure of the species evenness of a community), there was no difference in alpha diversity indices between cows and goats (Figure 1). The results of the alpha diversity assessment are shown in Appendix A. 

The Beta diversity, which assesses the difference among the bacterial communities, was determined using different algorithms. A Principal Coordinate Analysis (PCoA) plot based on weighted (Figure 2), and unweighted UniFrac distance matrix (Appendix A) shows that cows clustered separately from goats. Statistical analysis documented a significant difference between the studied groups of ruminant species. However, the results were influenced by high intergroup variability. PERMANOVA and PERMDISP results are listed in Table 1. These results indicate that the host species significantly affects the feces’ bacterial diversity. 

### 3.2. Taxonomical Composition

In the whole dataset, a total of 15 phyla (including 287 bacterial phylotypes) were detected, but only 8 of them, including Firmicutes, Bacteroidetes, Proteobacteria, Spirochaetes, Actinobacteria, Tenericutes, Patescibacteria, and Lentisphaerae, had a meaningful relative abundance (>0.5%). The abundances of phyla Epsilonbacteraeota, Planctomycetes, Elusimicrobia, Cyanobacteria, Kiritimatiellaeota, Fibrobacteres, and Fusobacteria were low (≤0.3%) and they are summarized as “others” in Figure 3. 

Firmicutes were detected as the dominant phylum in both groups of ruminants (75 ± 4.3% in goats and 74.9 ± 2.5% in cows). Regardless of animal species, the major order of Firmicutes was Clostridiales, which was represented at the family level mainly by Ruminococcaceae, Christensenellaceae, and Lachnospiraceae, followed by the less abundant Peptostreptococcaceae and Family XIII. The second most abundant phylum resulted from Bacteroidetes (17.7 ± 2.5% in goats and 19.5 ± 1.9% in cows), which was represented in both groups mainly by the order Bacteroidales with the families Rikenellaceae, Prevotellaceae, and uncultured Bacteroidales RF16 group and p-2534-18B5 gut group. Less abundant phyla Proteobacteria and Spirochaetes were represented mainly by the family Desulfovibrionaceae (class Deltaproteobacteria) and Spirochaetaceae (class Spirochaetes), respectively.

At genus level, the most abundant genera were the *Christensenellaceae* R-7 group, *Ruminococcaceae* UCG-005 and *Ruminococcaceae* UCG-010. Less abundant genera with a relative abundance higher than 1% were *Alistipes*, *Romboutsia*, *Rikenellaceae* RC9 gut group, *Eubacterium coprostanoligenes* group, *Family XIII* AD3011 group, *Ruminococcaceae* UCG-013, *Ruminococcaceae* UCG-014, *Ruminococcaceae* UCG-009, *Treponema* 2, *Solibacillus*, *Prevotellaceae* UCG-003, *Paeniclostridium*, *Prevotellaceae* UCG-004, and *Ruminococcaceae* NK4A214 *group*. Some of the sequences (22.6% in goats and 26.1% in cows) were not classified at the genus level and the lowest taxonomic assignment was achieved at the family or even order level. The most abundant unclassified genera were found in the order Bacteroidales and the families Ruminococcaceae, Lachnospiraceae, Bacteroidales RF16 group, Clostridiales vadinBB60 group, and p-2534-18B5. The relative abundance of taxa at different taxonomic levels is shown in Figure 3 and listed in the Appendix A. Bacterial genera with low relative abundance are summarized as "others" in Figure 3 and listed in Appendix A.

### 3.3. Determination of Taxonomic Biomarkers

To elucidate the differences in the composition of the microbiome of cows and goats, the linear discriminant analysis (LDA) with effect of size (LEfSe) was performed to determine the bacterial taxa with significantly different abundances. A total of 36 taxa with significantly different abundances were identified in the two groups of animals (Figure 4). Twenty-four taxa had significantly higher relative abundance in the goat group (green bars), and 12 taxa had significantly higher relative abundance in the cow group (red bars). Based on the multitaxonomic LEfSe analysis, at the phylum level, Bacteroidetes and Spirochaetes were enriched in goats. At the class level, Clostridia, Bacteroidia, and Spirochaetia were enriched in goats, whereas Erysipelotrichia and Bacilli were enriched in cows. At order level, Bacteroidales and Spirochaetales had significantly higher relative abundances in goats, while Erysipelotrichales and Bacillales had significantly higher relative abundances in cows. At the family level, higher abundances of Christensenellaceae, Lachnospiraceae, Ruminococcaceae, Spirochaetaceae, and Bacillaceae were found in goats, while Erysipelotrichaceae was enriched in cows. At the genus level, 12 taxa, including *Ruminococcaceae* UCG-005, *Christensenellaceae* R-7 group, *Ruminococcaceae* UCG-010, *Ruminococcaceae* UCG-009, *Ruminococcaceae* UCG-013, *Ruminococcaceae* UCG-014, *Ruminococcus* 1, *Ruminococcaceae* UCG-002, *Lachnospiraceae* NK4A136 group, *Treponema* 2, *Lachnospiraceae* AC2044 group, and *Bacillus* were significantly more abundant in goats, and 7 taxa, including *Turicibacter*, *Solibacillus*, *Alloprevotella*, *Prevotellaceae* UCG-001, *Negativibacillus*, *Lachnospiraceae* UCG-006, and *Eubacterium hallii* group were enriched in cows. All differentially abundant taxa are shown in the histogram (a) and cladogram (b) in Figure 4. 

## 4. Discussion 

In this study, we investigated the bacterial community in the feces of two ruminant species offered the same diet and kept under the same animal husbandry conditions to elucidate the influence of host species on microbiota structure and diversity. In general, the bacterial community composition in the fecal samples was dominated by Firmicutes and Bacteroidetes, the two most abundant phyla known to prevail in all ruminants. The prevalence of Firmicutes has been found in many studies on the bacterial composition in the feces of cattle and goats regardless of the type of diet [11,37,50,51,52,53], which is in agreement with our study. However, the ratio of Firmicutes/Bacteroidetes in the fecal populations has been associated with changes in the animal’s diet [34,51]. At genus level, the most abundant genera were the *Christensenellaceae* R-7 group, *Ruminococcaceae* UCG-005, and *Ruminococcaceae* UCG-010. They represented together 35.2% of the sequences in goats and 32.6% in cows. This is in good agreement with the findings of Andrade et al., who examined the fecal microbial populations in Nelore cattle and found that 16% of the sequences belonged to *Ruminococcaceae* UCG-005 and UCG-010 [54]. These two genera were also described for cattle, goat kids, and musk deer fecal microbiomes [55,56,57]. Less abundant genera were *Alistipes*, *Romboutsia*, *Rikenellaceae* RC9 gut group, *Eubacterium coprostanoligenes* group, *Family XIII* AD3011 group, *Ruminococcaceae* UCG-013, *Ruminococcaceae* UCG-014, *Ruminococcaceae* UCG-009, *Treponema* 2, *Solibacillus*, *Prevotellaceae* UCG-003, *Paeniclostridium*, *Prevotellaceae* UCG-004, and *Ruminococcaceae* NK4A214 group. Most of these bacteria are detected in the fecal microbiome of ruminants [32,54].

Regarding the influence of the host species on the fecal microbiome, the uniformity of individual distribution of bacteria in the community was not significantly different between cows and goats. The bacterial phylogenetic diversity was also not affected by the host species. On the other hand, the animal species had an effect on the count of individual bacterial species, resulting in significantly higher species evenness in cows compared to goats. These results indicate that species richness and phylogenetic composition are not affected by the host animals, while the equity in bacterial species abundance differs between cows and goats. The calculation of the distance between the studied groups resulted in a separation between goats and cows that was statistically significant despite the intragroup dispersion of the samples. All the algorithms used (Bray-Curtis, Jaccard, weighted, and unweighted UniFrac) produced similar results, indicating significant differences between cows and goats regardless of whether qualitative or quantitative measures of differences among communities were considered and regardless of whether phylogenetic relationships among features were included or excluded. However, the difference between the two groups of hosts was greater for the weighted UniFrac distance (55.1% on axis 1) than for the unweighted UniFrac distance (27.8% on axis 1). The unweighted analysis only considers the presence/absence of taxa, whereas the weighted analysis further evaluates the relative abundance of specific bacteria. This indicates that the relative abundance of certain bacteria contributed to the bacterial community distance between cows and goats. The LEfSE analysis identified taxa with significantly different abundances in each animal group. Mainly the *Ruminococcaceae* genera (UCG-002, UCG-005, UCG-009, UCG-010, UCG-013, and UCG-014) and *Ruminococcus* 1 were enriched in goats’ feces. *Ruminococci* are important bacterial species for ruminants due to their cellulolytic activity and ability to convert complex polysaccharides into a variety of nutrients for the host [58]. Their presence in the lower gut is crucial for efficient post-ruminal fiber utilisation [59]. This is confirmed by recent studies reporting that the utilisation of starch in the small and large intestines was mostly attributed to microbial fermentation rather than to host enzymes [60,61]. Moreover, a higher abundance of Ruminococcaceae sequences was found in the fecal samples of forage-fed animals than grain-fed animals [34,51].

We can consider it a positive result of our study that we did not detect potentially pathogenic or opportunistic microbes such as *Campylobacter*, *Salmonella*, *Bergeriella, Escherichia* or taxa of *Neisseriaceae*. These strains have been found by several researchers in both cows [50,62] and goats [63]. The occurrence of opportunistic bacteria in the aforementioned studies may be associated with the feeding of a high-grain diet, the negative effects of which on the host have been described in an increasing number of publications [64,65,66,67].

The influence of the host animal on the fecal microbiome has been investigated in a few studies [43,55,63]. Shabana et al. [63] compared the bacterial composition in the feces of sheep and goats of the same age, located on the same farm and offered the same diet (pellet feed and alfalfa hay). In contrast to our findings, alpha diversity was significantly different between the two animals, showing a higher species complexity in sheep compared to goats. Moreover, Ming et al. [43] analyzed fecal microbial communities in cattle and three populations of Bactrian camels. The results revealed significant distances in bacterial community structure among the four animal groups using the weighted UniFrac distance algorithm. This suggests a relationship between the relative abundance of bacteria and host species, which was also evident in our analysis.

There are many differences between cows and goats, not only in body size, rumen size, and rumen content passage rate, but also in feeding behavior, feed intake, digestive function, nutrient utilization, water economy, turnover rate, and digestive efficiency [68,69,70]. Some of these differences are innate, while others result from their adaptation and interaction with various environmental factors. In general, cows are typical grazers [41], while goats are known to be browsers and mixed feeders [42]. They have different abilities to utilize plant resources, resulting in different productive responses. Cows have a higher intake and digestive ability than small ruminants due to their larger intestinal capacity [14]. However, on high fiber, low quality forages, goats have better digestive efficiency than other ruminants, and one of the main reasons for this is the longer mean retention time of digesta in the rumen. All these factors certainly affect the rumen ecosystem and bacterial diversity.

The microbial community plays an important role in the overall nutritional and health status of the host. It is strongly affected by diet, behavior, eating habits, and animal management practices [63]. In the present study, cows and goats were kept in the same location and fed the same diet. As a result, their fecal microbiota composition and diversity were noticeably similar.

## 5. Conclusions

This study demonstrated that the investigation of the fecal microbial ecosystem in ruminants can help to understand the effects of diet and host species on bacteria involved in feed digestion. The results presented here revealed that the fecal bacterial community of two ruminant species fed the same high-fiber diet were largely similar. However, the influence of the host animal was significant. The differences were mostly caused by the different abundance of some bacterial taxa, which affected species evenness, while richness and diversity were not significantly influenced by the animal species. 

## Figures and Tables

**Figure 1 animals-12-02011-f001:**
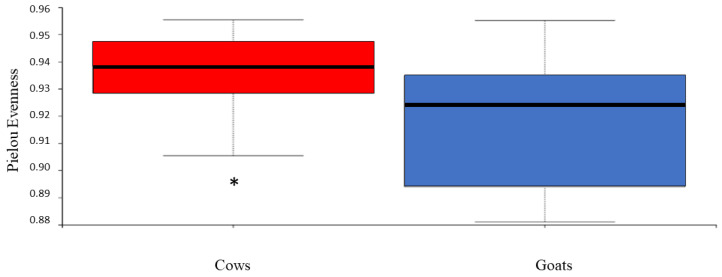
Boxplot of evenness values (Pielou’s index) for 16S rRNA gene sequences retrieved from the feces of cows and goats. ***** indicates a significant difference (*p* < 0.05).

**Figure 2 animals-12-02011-f002:**
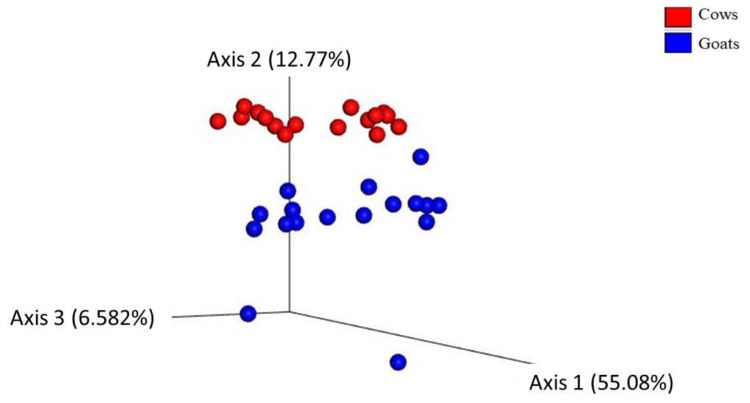
Principal Coordinate Analysis (PCoA) showing the weighted UniFrac distance matrix of bacterial 16S rRNA amplicons from fecal samples of cow (red color) and goat (blue color) groups. Each dot represents one sample. The percentage of variation explained by the plotted principal coordinates is indicated on the axes.

**Figure 3 animals-12-02011-f003:**
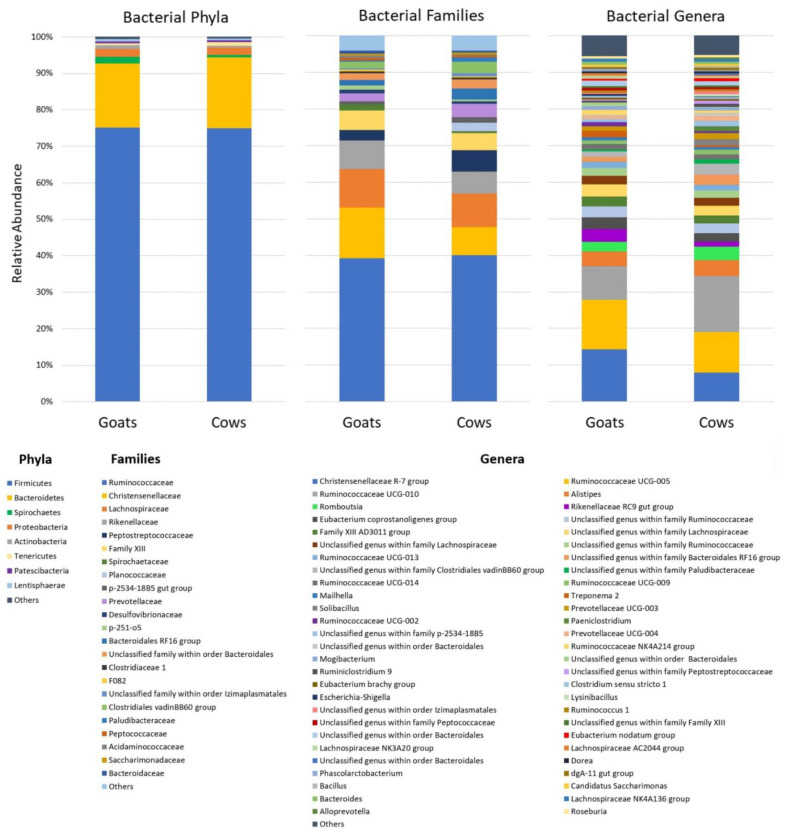
A comparison of the fecal bacteria of goats and cows at several taxonomic levels. The relative abundance is illustrated at the phylum, family, and genus level. Taxa with a relative abundance of less than 0.5% are grouped as “Others”.

**Figure 4 animals-12-02011-f004:**
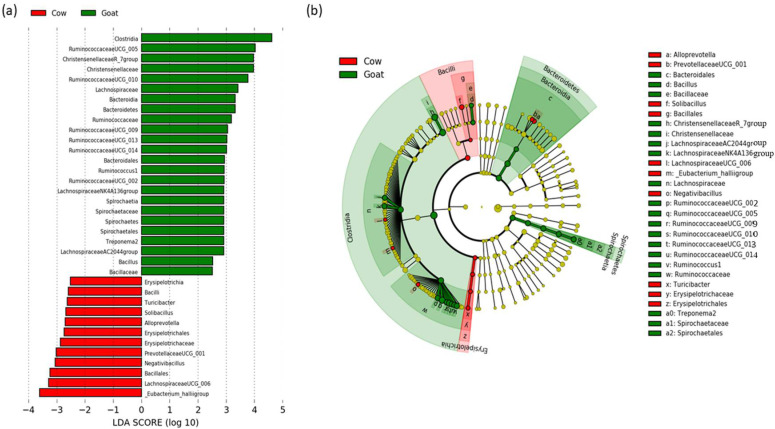
Linear discriminant analysis (LDA) scores for 36 bacterial phylotypes with significantly different abundances in cow and goat fecal samples. (**a**) The bar length represents the log10-transformed LDA score indicated by the vertical dotted lines. Negative (red bars) LDA scores represent bacterial taxa overabundant in cows, while positive (green bars) bacterial taxa are overabundant in goats. (**b**) Cladogram showing the differences in enriched taxa in cows (red) and goats (green).

**Table 1 animals-12-02011-t001:** Permutational multivariate analysis of variance (PERMANOVA) and dispersions (PERMDISP) showing significant differences in beta diversity between cows and goats (*p* < 0.05).

	PERMANOVA *p*-Value (* *p* < 0.05)	PERMDISP *p*-Value (* *p* < 0.05)
Bray curtis distance	0.001 **	0.01 *
Jaccard distance	0.001 **	0.002 *
Weighted unifrac distance	0.001 **	0.02 *
Unweighted unifrac distance	0.046 *	0.04 *

* Significant difference (*p* < 0.05). ** Strong significant difference (*p* ≤ 0.001).

## Data Availability

The data presented in this study are openly available in the Sequence Read Archive under the accession number PRJNA826341.

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
