# Peer review of "Host Species Affects Bacterial Evenness, but Not Diversity: Comparison of Fecal Bacteria of Cows and Goats Offered the Same Diet"

_animals, 2022, doi:10.3390/ani12162011_

Round 1

Reviewer 1 Report

Before final acceptance, the authors need to look more thoroughly in the international literature, as there are some recent references which are relevant and will help in the discussion section. So, please read the literature and improve the discussion.

Then, the manuscript can be accepted.

Author Response

We express our gratitude to the reviewers for their time spent to read and comment on our manuscript. Kindly, find attached the answers to each doubt and comment. All the changes have been made accordingly to the suggestions. We hope that the revised manuscript results now sufficiently improved to deserve the publication.

Reviewer 2 Report

Review report for manuscript ID: Animals-1742411

Host species affects bacterial evenness, but not diversity: comparison of fecal bacteria of cows and goats fed the same diet

The aim of this paper was to determine differences in the faecal bacterial community of cows and goats fed the same pasture based diet. The authors show that there are no major differences in the alpha diversity indices between the two ruminant species (with the exception of Pielou’s evenness). The community composition of the cows and goats cluster separately, however there is considerable variation between the animals within a species. The paper presents good methodology and the results are clear and easy to follow. Please see specific comments on each section below where I have highlighted suggestion to improve the manuscript/areas of weakness/concern.

Introduction:

Introduction gives a good background for the rumen and the rumen microbiome but the samples are collected from faeces. The introduction would benefit from a change in focus to instead discuss the faecal microbiome specifically and what we know on this so far (not the rumen) – there are papers that compare faecal samples to the rumen that could be introduced here (e.g. doi: 10.3390/ani9080498). The introduction should provide some information on why faecal samples were used and what this tells us – is there evidence to show they are a good proxy for the rumen (if that is the point of the study?) Is it because they were non-invasive samples. The hypothesis also mentions that differences might be observed because of the differences in feeding habits and physiology of the two ruminant species. It would be beneficial to add a short paragraph summarising these differences to benefit your reader (you have some of this in the discussion). You could include a brief summary of the introduction you have so far e.g. lines 48-52 with all the factors that affect the rumen microbiome and then move on to talk about the faecal microbiome specifically after this.

Methods:

Methods provide adequate detail to summarise what was done. I have some specific questions below:

Line 95 – Were all samples collected on the same day? If so, how long between sample 1 and 30?
Line 96 – how were they transported? Warm/on ice? Were they stored at all before freeze drying? 
Line 159 – 167 – was age/lactation/pregnancy status accounted for in your models? There is quite a lot of variation, particularly in age in your sample population. Would be good to see if these account for any of the variation in your dataset

Results:

Figures and tables presented here and in the SM are clear and easy to understand. For your beta diversity results, I would suggest moving table S3 into the main text as this contains important information relevant to the paper. At the moment, you don’t provide a p-value in the text for B-diversity. It is also important to show to your reader the results of the B-dispersion. As this is significant, you can’t say for definite whether the difference you’re seeing is due to differences in community composition or whether this is due to variation within your groups.

I would also like to see the relative abundance of the genera that you have plotted in Figure 3 in your supplementary material.

I would strongly recommend adding lactation stage and age into your models as it is known that the faecal community changes with both age and stage of lactation – this may help to explain some of the variation between your groups.

Line 170-177 consider rewording – “With the exception of Pielou’s evenness (a measure of species…), there was no difference in alpha diversity indices between cows and goats”.

Line 175-177 move to discussion

Table S2 – Please add error values to this table (standard deviation/error).

Line 199 – what is a meaningful relative abundance?

Line 203-212 – Provide some quantitative information in this paragraph, particularly for Fimicutes and Bacteroidetes as a minimum i.e. what % of reads were Firmicutes/Bacteroidetes for cows and goats.

Line 213: This seems quite a high proportion of unclassified reads.

Discussion:

Why do you think that so many of the genera were unclassified? Can you provide a reason for this? Perhaps the choice of reference database? Greengenes has not been updated in a long time and I believe it is no longer being maintained.

Similarly to the introduction, I feel that there is too much comparison with rumen studies (e.g. line 301 to 306). You are going to see differences compared to the rumen as your samples are not from here. Your manuscript would strongly benefit from a change in angle here to focus on the faecal community specifically (there are lots of papers to compare to). The comparisons in the discussion should be with other papers that have looked at faecal samples from cows and goats first and foremost before you talk about the rumen.

Line 326 – 330 – Function in the rumen is explained, what is the function in the hindgut?

Line 333 – you don’t have rumen samples

Somewhere in your paper, you need to specify why you are using faecal samples – are you expecting these to reflect the rumen community? If so, you should check the literature (and cite it) to show that this can/can’t be done. There are papers such as https://doi.org/10.1016/j.animal.2021.100281 which shows large difference in the rumen and faecal samples from the same animals. The discussion should be re-focused specifically around the hind gut, not compared with the rumen without evidence that it can be used as a proxy. Microbes that leave the rumen act as a source of protein for the animal (they are digested). It does not necessarily follow that these microbes will seed the hind gut as the nutrients reaching this part of the digestive tract will be different to that of the rumen itself. Your comparisons should be only with other papers that have looked at the faecal community. You could say that others have looked at difference in rumen populations between different ruminant species and have found a difference – but this shouldn’t be the main focus.

Line 370 – 383 – Whilst interesting, this is not directly relevant to the study. Consider deleting.

Line 384 – 388 – consider deleting and re-write this. Too much focus on the rumen here this is not what you looked at in your study.  

Line 391 – 392 – consider re-wording “bacteria involved in feed-fermentation” to “feed digestion”

References:

The references included are up to date and include a good range but are largely focussed on the rumen and not the faecal bacterial community.

General comment:

This is a well presented manuscript, methods are clear and results are well written. The introduction and discussion, however are unclear in their focus and I suggest re-focusing these. Comparison between your communities in this study and those in the rumen should be minimal. A change of focus will improve the story you are trying to tell. The "big-picture" of the study is not clear at the moment. 

Author Response

(The authors gave the same response as above.)

Reviewer 3 Report

  1. The publication deals with an interesting aspect concerning the aspects of ecological use of goats and cattle. The research would have been fully developed if sheep had been introduced
  2. For research purposes, it is necessary to clearly describe what research results are relevant to the breeding practice
  3. The material and methods of line 87-101 need major revision

- it should be completed at what age were the animals

- how long they stayed in the pastures and how many stayed in the stables

- accurately describe the feeding system

- whether the animals lived on the same pastures together

- how the animals are dewormed on the farm.

- describe the botanical composition of the sward

The summary does not correspond to the assumed hypotheses, it should be clarified

Author Response

(The authors gave the same response as above.)

Round 2

Reviewer 2 Report

Dear Authors, 

Thank you for taking the time to take on board the suggestions made in the first round of review. The manuscript is much improved. 

Author Response

We would like to thank you for revising our paper. We believe that your comments and suggestions were essential for the improvement of our manuscript.

Thank you again.